# Manufacturing and Characterization of Coatings from Polyamide Powders Functionalized with Nanosilica

**DOI:** 10.3390/polym12102298

**Published:** 2020-10-08

**Authors:** Maria Fernández-Álvarez, Francisco Velasco, Asuncion Bautista, Flavia Cristina M. Lobo, Emanuel M. Fernandes, Rui L. Reis

**Affiliations:** 1Department of Materials Science and Engineering, IAAB, Universidad Carlos III de Madrid, Avda. Universidad 30, Leganés, 28903 Madrid, Spain; mfalvare@ing.uc3m.es (M.F.-Á.); mbautist@ing.uc3m.es (A.B.); 23B’s Research Group, I3Bs—Research Institute on Biomaterials, Biodegradables and Biomimetics, University of Minho, Headquarters of the European Institute of Excellence on Tissue Engineering and Regenerative Medicine, AvePark, Parque de Ciência e Tecnologia, Zona Industrial da Gandra, 4805-017 Barco, Guimarães, Portugal; flavialobo@i3bs.uminho.pt (F.C.M.L.); rgreis@i3bs.uminho.pt (R.L.R.); 3ICVS/3B’s—PT Government Associate Laboratory, 4710-057 Braga/Guimarães, Portugal

**Keywords:** polyamide, powder coating, silica nanoparticles, xenon weathering, mechanical properties, wear resistance

## Abstract

Polyamide coatings are thermoplastics with great advantages such as a good corrosion protection of the base metal and wear resistance. Their application as powder coatings is an environmentally friendly option that is currently attracting growing interest. However, during their life service, they can sometimes be exposed to conditions that they are unable to stand. In this work, a polyamide 11 (PA11) powder was reinforced with different percentages of silica nanoparticles (1–3 wt. %). Powder mixtures were prepared through extrusion followed by compression molding processes to manufacture coatings. For the coatings under study, the effect of 500 h xenon exposure was studied in order to know their ultraviolet (UV) resistance. Attenuated total reflection-Fourier transform infrared spectroscopy (FTIR-ATR) and differential scanning calorimetry (DSC) tests were performed to study changes in polymer structure and if they are affected by nanoparticles. The effect of nanoadditions and xenon exposure on hardness and stiffness were also evaluated. Furthermore, reciprocal wear tests were performed before and after irradiation, and the wear tracks were analyzed using optoelectronic microscopy and scanning electron microscopy (SEM). Finally, the aesthetic properties were measured. The results reveal improvements in mechanical and wear properties when 1% nanosilica is added to the PA11, which then become more relevant after xenon radiation exposure.

## 1. Introduction

In recent years, the great advance of nanotechnology has allowed the design and development of new materials with improved properties [1], including functional polymeric coatings [2]. The manufacturing of more resistant organic coatings is important for extending their service life and, therefore, avoiding corrosion of the base metal [3,4]. The development of hybrid nanomaterials (inorganic–organic) has allowed the manufacturing of organic coatings with improved mechanical properties and wear resistance, which can keep their barrier properties for longer times in aggressive environments.

Most of the studies based on modifying organic coatings with nanoadditons have been carried out for liquid coatings, especially for thermoset coatings [5,6,7,8], although, in the literature, there is already some published research regarding thermoplastic liquid coatings modified with nanoparticles [9,10]. This is related to the fact that, generally, thermosets are used for functional coatings, because of their mechanical properties and their excellent adhesion to the substrate [11,12,13]. Nevertheless, thermoplastic coatings have other advantages, such as better finishing and easy reparation in case of failure [14].

On the other hand, the use of organic powder coatings is growing due to their advantages over conventional liquid coatings [15]. Their main benefit is their great environmental advantage because they are free of solvents [14,16]. Furthermore, other advantages of powder coatings are the possibility of recovering the powder that has not adhered during the painting process, and their better surface finishing and energy savings compared to liquid coatings [17,18]. Thermoset powder coatings are the most common in the market and have been more studied to date. However, thermoplastic coatings were the first to be applied as powder coatings [19] and have a clear interest.

The most common commercial thermoplastic powder for coatings in the market is polyamide. In this work, a polyamide 11 (PA11, also called nylon 11) powder was selected because of its advantages, such as its natural origin (from castor oil, a 100% renewable resource), its high impact resistance, low water absorption and corrosion, chemical and wear resistance [20]. PA11 also offers good performance in outdoor applications [21] and it has been used in long-term applications in the automotive or construction sectors and in piping applications [22]. There have been previous attempts to improve their properties using nanotechnology.

Among the different types of nanoreinforcements used in organic coatings, there are many previous studies showing that silica nanoparticles can be a useful strategy to improve the different properties of organic coatings [23]. Silica nanoparticles are non-toxic, optically transparent and chemically inert [24]. For powder coatings, there are studies related to the functionalization with silica nanoparticles of thermoset resins on mechanical properties [14,25,26] in order to obtain the desired surface properties. However, for thermoplastic powder coatings, there is practically no published literature about the effect of nanoparticles on their properties, with only one study found, where Hedayati et al. [27] managed to improve the mechanical properties of a polyether–ether–ketone (PEEK) with silica nanoparticles.

There are some previous results related to the functionalization of conventional polyamide nanocomposites with silica. García et al. [28] improved the elasticity of a polyamide 6 nanocomposite using silica nanoparticles. Moreover, Petrovicova et al. [29] managed to enhance the wear resistance of a PA11 nanocomposite. For this reason, the use of this type of nanoparticles in polyamide, manufactured as a coating, is considered as the object of study in this article.

In addition, the mixing method must be carefully considered to obtain a homogeneous dispersion of the nanoreinforcements in the coating. A bad distribution of the nanoparticles could cause negative effects, since agglomerates could act as stress concentration points, weakening the organic coating and even decreasing its barrier effect [30,31]. For this reason, in this work, extrusion has been employed as a mixing method, since it can obtain homogeneous polyamide materials (coatings and composites) with different nanoreinforcements [32]. In situ polymerization method also leads to an excellent distribution of nanoparticles [33], but it cannot be considered as the work is focused on the functionalization of a commercial powder coating.

This work aims to enhance the advantages of powder coatings, in this case using a green alternative such as PA11, and adding silica nanoparticles in order to improve its mechanical properties and wear resistance. Moreover, any outdoor exposure can induce photodegradation of polymer materials [34], and nanoadditions can influence the ultraviolet resistance of polymers, positively [35] or negatively [34]. For these reasons, in this work, a study has been also included to see how exposure to a xenon lamp influences these PA11-reinforced coatings.

## 2. Materials and Methods

### 2.1. Materials and Manufacturing of PA11 Coatings

In this work, a white PA11 powder, provided by Rilsan^®^ (Arkema, Colombes, France), was employed (D50: 30 µm). As a nanoreinforcement, hydrophilic Aerosil^®^ 90 silica nanoparticles were used (Evonik, Essen, Germany). The surface area of the silica nanoparticles was 75–105 m^2^·g^−1^ and the average size 20 nm. Both PA11 powders and nanosilica particles were supplied and stored under dry conditions. In this work, low amounts of nanoparticles were used due to the improvements that have been achieved in other types of coatings using these concentrations [25,26]. The percentages of SiO_2_ added to the PA11 powder coating were 1, 2 and 3 wt. %.

The extrusion process was carried out to mix the two materials (PA11 and nanoparticles). The raw materials were pre-mixed and then fed to a Rondol SCF modular co-rotating twin-screw extruder, with a screw diameter of 16 mm and a length to diameter ratio (L/D) of 25, and equipped with a die diameter of 3 mm. The mixture was placed in the hopper and automatically fed at a constant rate with a volumetric dosing unit (Shini Plastics Technologies, Pune, India). The temperature profile along the barrel was set from 170 to 210 °C, with the screw speed at 50 rpm. The extruded material was cooled in a water bath and subsequently ground with a Scheer pelletizer (Maag group, Oberglatt, Switzerland) to produce composite pellets suitable for compression molding. Finally, the produced pellets were dried in a vacuum oven at 60 °C for moisture stabilizing. The extrusion and pelletizing processes were carried out not only for the three SiO_2_-containing materials (labelled as 1%, 2% and 3%), but also for the neat PA11 (labelled as 0%), to properly compare their properties. As the first stage of the research, where the effect of the nanoadditions is the objective of the study, the cryogenic milling of the pellets for an electrostatic application was discarded because of its cost, and compression molding was chosen instead. The coatings were applied once extruded and grounded (in the form of pellets), in order to optimize the necessary amount of silica nanoparticles to improve the properties of the PA11. 

The compression molding process of the obtained PA11-based pellets was performed using a hydraulic press (Carver, Wabash, IN, USA) to produce the coatings of the four compositions.

A fixed amount of pellets from each composition was collected and dispersed between aluminum (70 × 40 × 0.8 mm^3^) and Teflon sheets. This system was placed in a hydraulic press at a temperature of 220 °C for 6 min to allow the melting and homogenization of the pellets. The pellets were further pressed for 2 min at a pressure of 3.5 MPa, followed by a cooling stage of 5 to 10 min to near room temperature using water and maintaining the pressure. Finally, the pressure was removed, allowing the unmolding of different PA11-based coatings. The thickness of the organic coatings was about 122 ± 12 µm.

### 2.2. Characterization and Testing

In order to study the presence of the silica nanoparticles and the efficiency of the extrusion process, a thermogravimetry analysis (TGA) of the non-exposed coatings was performed in air in a SETSYS Evolution instrument (Setaram, Caluire-et-Cuire, France), using a heating rate of 20 °C·min^−1^.

An accelerated weathering test was carried out using a xenon lamp chamber Solarbox3000 (Cofomegra, Milano, Italy). The four PA11-based coatings under study were tested. A nominal irradiance of 550 W/m^2^ (real irradiance of 537 W/m^2^) was applied in the range of 290–800 nm, with 500 h of exposure with a relative humidity of 50% and a black panel temperature of 65 °C. The distance between the lamp and sample was 23.5 cm. All the characterization tests were carried out before and after exposure to xenon lamp, and will be referred to hereafter as 0 and 500 h, respectively.

The structure of PA11-based coatings was analyzed by Fourier transform infrared spectroscopy (FTIR), using a Bruker Tensor 27 spectrometer (Bruker, Billerica, MA, USA) with a Golden-Gate-attenuated total reflection (ATR) accessory (Specac, Orpington, UK). The FTIR-ATR spectra of the coatings were recorded in the range of 4000–600 cm^−1^ at a resolution of 4 cm^−1^ and 32 scans.

Differential scanning calorimetry (DSC) experiments were performed in a model 822 (Mettler Toledo GmbH, Greifensee, Switzerland), using a refrigerated cooling system and nitrogen as a purge gas (flux gas 80 mL·min^−1^). Samples of 8.4 ± 0.3 mg were used and an effort was made to keep to a similar geometry in order to assure the same thermal resistance. All the experiments were performed at 20 °C·min^−1^, starting from 0 to 250 °C. Two scans were performed to check that there were no significant changes between them due to the thermal history of the polymer. The melting temperature (*T*_m_), melting enthalpy (Δ*H*_m_), crystallinity (*X*_C_) and the glass transition temperature (*T*_g_) were analyzed. In order to calculate the *X*_C_, the following formula was used*X*_C_ (%) = (Δ*H*_m_/Δ*H*_0_·*w*)·100(1)where Δ*H*_0_ is the enthalpy corresponding to the melting of a pure crystalline PA11 (226.4 J·g^−1^, according to [36]) and *w* is the mass fraction of PA11 in the coating [37], measured by thermogravimetry.

The surface morphology of the coatings was studied by scanning electron microscopy (SEM), using Teneo-LoVac (Thermo Fisher Scientific Inc., Waltham, MA, USA) equipment, under a 10 kV electron beam. Semi-quantitative analyses were also performed with an energy-dispersive spectroscopy (EDS) analyzer (at 15 kV).

In order to study the mechanical properties of the PA11 coatings, stiffness and universal hardness (HU) were measured [38]. A ZHU 2.5 universal hardnessmeter was used (Zwick Roell, Ulm, Germany). The applied load was 5 N, the speed of load application was 0.5 mm·min^−1^ and the speed of load removal was 2 mm·min^−1^. At least seven measurements were performed for each organic coating.

Sliding wear determination was carried out using a UMT Tribolab (Bruker, Billerica, MA, USA) reciprocating tribometer. The countermaterial used was a stainless steel ball with a diameter of 6 mm and tests were made on the coated aluminum sheets. The conditions were 15 N, 15 Hz and a length of the wear track of 5 mm. The tests were performed for 40 min and under dry conditions. Four tests were made for each organic coating. The depth of the wear tracks was measured in an opto-digitalOlympus DSX500 (Olympus Corp., Tokyo, Japan) microscope. At least nine measurements were done on each wear track. The wear mechanism was also studied by SEM.

The aesthetic properties (color and gloss) were also studied. The color of the organic coatings was measured under D65 illuminant, 2° observer and exclusion of the specular component (SCE) on X-Rite ColorEye^®^ XHT (X-Rite Inc., Gran Rapids, MI, USA), in accordance with ISO 11664 standard. Gloss measurements were performed with a Refo 3 glossmeter (Hach Lange, Loveland, CO, USA), using 60°, based on ISO 2813 standard. Four measurements were done in each case.

## 3. Results

TGA analysis of the non-exposed coatings is shown in Figure 1. Weight loss is consistent with the manufactured coatings, and the amount of PA11 in each coating (*w*) can be measured. For 0% coating, it corresponds to a quantity of 70%, being the rest fillers and pigments of the powder raw material. For the rest of coatings, *w* was 69% (for 1%), 68% (for 2%) and 67% (for 3%).

In Figure 2, it can be observed the FTIR-ATR spectra obtained for all PA11 coatings at 0 h (Figure 2a) and 500 h (Figure 2b) of xenon exposure. Besides, the main peaks are marked in Figure 2a and identified in Table 1.

There are no great differences among the FTIR-ATR spectra obtained for all PA11 coatings, neither at 0 h nor after 500 h of xenon exposure. Before exposure, all the typical bands of polyamide are found. The main difference among coatings relates to the band lying at 1114 cm^−1^, corresponding to the Si-O-Si bond of silica nanoparticles, that increases when their amount in the PA11 coating does [24].

After 500 h exposure, there are no meaningful structural changes that could imply a noticeable deterioration due to their exposure to solar radiations. Kaci et al. [39] suggest that UV exposure of PA11 promotes the appearance of imide photooxidation products, being absorption bands located at 1736 and 1690 cm^−1^, and the formation of carboxylic acid groups (band located at 1713 cm^−1^). The FTIR of developed coatings (Figure 2c) suggests that these phenomena cannot be discarded in the PA11 coatings, although longer exposure times would be required to completely assure it. The band at 1734 cm^−1^ can be slightly detected at 0 h (Figure 2c), suggesting their presence in the original coatings.

Thermal studies are important to verify whether nanoparticles are influencing or not the crystallinity of thermoplastic coatings before and after weathering, which is mainly related to their mechanical properties [42]. In Figure 3, the DSC curves are shown for non-exposed and the xenon exposed PA11 coatings (0% and 3% as an example). In the graph, the two main parts to be analyzed in the thermograms have been marked with rectangles. The red and small rectangle corresponds to the *T*_g_ of the amorphous part of PA11, while the grey rectangle corresponds to the melting of the crystalline part of the polymer [42]. Table 2 and Table 3 collect the values of *T*_g_, the Δ*H*_m_ and the % of crystallinity (Equation (1)) of the four PA11 coatings before (0 h, Table 2) and after (500 h, Table 3) xenon exposure.

Regarding the *T*_g_ results, the values are similar for the four PA11 non-exposed coatings. Some studies observed that *T*_g_ can be increased by adding nanoreinforcements [43], but others did not find differences with the nanoadditions [22,29]. After PA11 coatings have been exposed to xenon lamp for 500 h, the *T*_g_ decreases for all coatings. Besides, it should be noted that the 0% coating is the material with the lowest *T*_g_ after the xenon exposure. As the destruction or formation of new types of bondings under xenon radiation only occur at a limited extent (Figure 2), the decrease in *T*_g_ after exposure to xenon may be linked to some chain scission [39], that in this case, is more pronounced in the 0% coating. Therefore, silica nanoparticles seem to partly balance the chain scission of the PA11 by new bond formation from the formed free radicals, maybe due to their already checked effect to enhance polymerization [11]. In any case, the chemical and structural processes are very complex and depend on several factors [44] and, in this work, they should not be large changes since they are not appreciated either by FTIR-ATR (Figure 2).

PA11 presents a crystallization event (Figure 3), but no noticeable effect was observed on *T*_m_ due to the addition of silica nanoparticles (Table 2 and Table 3). On the other hand, the Δ*H*_m_ of 0 h coatings seem to slight decrease when adding nanoparticles. The crystallinity is related to the Δ*H*_m_, so it also decreases. The 0% is the organic coating that has the highest melting enthalpy and, therefore, the highest crystallinity. The crystallinity can vary widely with the addition of nanoreinforcements to polymers, from studies that verify an increase in crystallinity [3], to those showing a decrease [45] or even studies where there is no variation in crystallinity [46]. In this study, the small differences between 0% and SiO_2_-containing coatings can suggest a small hindering on the alignment of PA chains to form crystals and lamellar organization of the chains of the polymer [45], but the effect can be considered almost negligible.

In the previous published literature, it can be seen that, after weathering exposure, polymers can also sometimes increase crystallinity (due to the loss of the amorphous part) [44] and in other cases, their crystallinity can decrease [47]. For the coatings under study, after their exposure to the xenon lamp for 500 h, their crystallinity is reduced in the same way as *T*_g_, as previously found [42]. The 0% is the polymer that loses more crystallinity, as happened with *T*_g_ value, while the nanoparticle containing coatings decrease their crystallinity with the xenon exposure at a lower extent, maintaining very similar values in all three cases. So, the order in the polymeric structure is decreased by the energy of the radiation, probably because the chain scissoring generates shorter chains. However, silica nanoparticles seem to slow down this phenomenon. This beneficial effect of silica nanoadditions against UVB weathering has previously been reported by other authors [48,49]. The nanosilica particles have been suggested to protect the matrix by absorbing UVB photons and converting them into heat [50] or by scattering the UVB rays [51].

The surface of all PA11 coatings under study were examined by SEM before and after xenon exposure. In Figure 4, examples of the micrographs obtained from the non-exposed 0% coating and the coating with the highest percentage of nanoparticles (3%) can be observed. EDS analysis were also carried out to identify the pigments and fillers contained in this PA11 commercial powder. The pigments and filler are observed as clearer spots and can be identified with TiO_2_, Al_2_O_3_ and AlPO_4_. The microscopic study of the non-exposed PA11 coatings shows no changes on the surface morphology or formation of irregularities due to nanoadditions, as the ones found in other systems [7,52]. Silica nanoparticles are not affecting negatively the final morphology of the PA11 coatings. Besides, it could be discarded that large agglomerates of silica addition have been formed, an important aspect for not losing mechanical properties [29,32].

Coatings after exposure to xenon lamp have also been examined (Figure 5). In all coatings, small imperfections were observed due to irradiation. Shallow microcracks are observed on all surfaces (marked with arrows), as well as small pores (marked with circles). At the microscale, all exposed PA11 coatings have this type of irregularities, although, in any case, the cracks seem very superficial, at least for 0% and 1% coatings. Moreover, though microcracks are seen in 1% coatings, micropores seem more difficult to observe than in the other studied materials and its surface could be considered less damaged by the radiation. The surfaces of 2% and 3% coatings appear more intensively damaged, indicating that for weathering applications the increase of concentration of silica nanoparticles impacts negatively in the coating performance. The observed formation of pores can be related to a chain scissoring process caused by the high energy of the UVB radiation. Other authors have suggested that this damage can be related to structural changes because of hydrolysis [53], but FTIR-ATR (Figure 2) has not shown this effect on tested PA11 coatings.

In Figure 6, it can be observed the mechanical properties measured for all PA11 coatings. A good mechanical resistance that stands external agents is very important for organic coatings. In this way, the coatings do not easily deteriorate, avoiding the corrosion of the base metal. Both the universal hardness (HU) of each coating and the stiffness were studied.

First, focusing on PA11 coatings not exposed to xenon lamp (0 h), it can be observed how both mechanical parameters increase when the amount of silica nanoparticles increases (especially for HU). In the case of thermoplastic materials, when nanoreinforcements are added, they act as a mechanical couplings with the polymeric matrix [54], being able to increase their rigidity and hardness. In addition to the interaction with nanoreinforcements, there are other factors such as crystallinity, *T*_g_ or morphology that could also affect the mechanical properties of the coatings [29]. As composite coatings have lower crystallinity than 0%, the increase of hardness due this fact can be discarded. In this case, silica nanoparticles must difficult the movement of polyamide chains under loading, increasing by this mechanism hardness and stiffness. On the other hand, the observed increase in hardness with nanosilica concentration and the small deviation of data also suggest an adequate homogenization of nanoparticles in the PA11 matrix and a good embedding [55].

Regarding the xenon irradiation exposed coatings (500 h), different behaviors are observed. In the case of the 0% and 1% PA11 coatings, the hardness and stiffness increase after exposure, while both parameters decrease for the 2% and 3%. The possible presence of imide groups (suggested by FTIR-ATR, Figure 2c) can be related to the increase of strength sometimes observed for the materials, despite crystallinity reductions (Table 2 and Table 3), while chain scissoring acts in the opposite direction, reducing mechanical properties. In the present work, it seems that a large number of silica nanoparticles (2% and 3%) are harmful for the weathering resistance, as images in Figure 5 also have suggested. Under UV, more chain scission takes in the polymers with higher amounts of nanoparticles, and therefore a decrease in the hardness of these coatings is observed. Hence, a high amount of nanoparticles can be an obstacle for chain recombination after scissoring.

With the aim of obtaining information about the performance of the four PA11 coatings and their durability under abrasive conditions, reciprocal wear tests were carried out on. The coefficient of friction (COF) during the wear test of each PA11 coating was monitored (Figure 7). The evolution of COF of all PA11 coatings is very similar to other previous studies [12,56]. In the graphs, two stages can be observed: the first stage, when the COF shows a fast increase, corresponds to the beginning of the wear, and the second stage takes place when the COF begins to remain almost constant (steady-state) in the test and coincide with the formation of a wear track and the progress of the abrasive wear in the organic coating. Regarding the final COF of the PA11 coatings, it is a value similar to that found in other studies with polyamides, as is the case of Li et al. [3]. Considering the results of the non-exposed PA11 coatings (Figure 7a), the COF evolution is very similar for all organic coatings. A small difference is observed for the 1% coating that seems to achieve the steady-state (with the same COF value of the other coatings) a little bit later than the other coatings. However, the delay on the onset of the formation of the wear track is clearly observed for 1% coating after 500 h exposure (Figure 7b). The 1% nanosilica addition increases the hardness of the coating after exposure (Figure 6a), fostering its wearing resistance during the first stage of the test. Wear resistance is usually related to mechanical properties such as the hardness and rigidity of the material. Nanoreinforcements often increase the hardness and rigidity of the matrix [55], improving the wear resistance [25,32]. However, an increase in hardness does not always lead to an enhancement of the wear resistance [57].

In Figure 8, the optoelectronic microscope images are shown for the exposed coatings (the images of the coatings at 0 h were quite similar). The green areas correspond with the intact coatings, while the blue area are the wear tracks. Within each wear track, those dark-purple areas would be the deepest ones, meaning greater wear. Taking into account the images, it is observed that, in general, the PA11 coatings seem to have similar wear, except in the case of the 1% coating, where a clear less depth is observed, since practically no dark-purple areas are observed. Therefore, the 1% reveals to have the best wear performance.

Once the images of the wear tracks were obtained, the average depth of each wear track was calculated from images such as those in Figure 8. These data are summarized in Figure 9. In the 0 h condition, the 1% is the coating with the best wear resistance, with an average depth of 69 µm, while the other three coatings have average depths of 79 µm, approximately. These results can be coherent with the small delay observed for the onset of the abrasive track on the material (Figure 7a).

For the coatings exposed to the xenon lamp (Figure 9), the 0%, 2% and 3% PA11 coatings show the same depth values than before the exposure. In this case, the decrease on the hardness because of the chain scission observed for 2% and 3% coatings (Figure 6a) does not cause a decrease on the wear resistance with exposure. However, in the case of irradiated 1%, its average depth value decreases by about 10 µm if it compared with that determined before the exposure. Hence, 1% coating increases its wear resistance with weathering unlike the other studied coatings. The increase on the hardness of the coating (Figure 6a) as well as the presence of a very reduced amount of micropores and defects in its surface (Figure 5) after irradiation can contribute to understand this result and explain the better performance of this coating.

In order to deepen the study of wear and to know the wear mechanism of the different PA11 coatings, the wear tracks of all the coatings were analyzed by SEM (Figure 10 and Figure 11). For all cases, cracks perpendicular to the movement direction of the countermaterial can be seen. The cracks seem wider for irradiated coatings, especially for the one without additions (0%). These results do not exactly show the same trend than neither the depth of the measured for the tracks (Figure 9) nor the hardness of the coatings (Figure 6a). However, the wear mechanism allows an easy understanding of this apparent discrepancy.

As can be seen in Figure 11, where some detailed images of the bottom of the wear track are shown, a two-step wear mechanism is shown. First, the sliding countermaterial tear pieces from the surface of the coating. Then, though some of them are lost as powdery residues, other remain on the wear track and subsequent slides of the pin plastically deform and stick them on the surface, partially covering the previously formed cracks or defects. Part of those debris are bonded again in the wear track. This kind of wear mechanism in a polyamide has already been observed [32,58]. The two level structure of the defects (the lowest initially eroded surface and the highest surface of redeposited material) can be seen. Materials with higher hardness suffer initially less material tearing, while very soft materials favor the plastic deformation and rebonding of the material.

The good wear performance observed for 1% coating after xenon exposure (Figure 9) can be understand because of its hardness (Figure 6a) that limits the amount of abrasively worn material. On the other way, the apparent non-harmful effect of the low hardness of 2% and 3% irradiated coatings (Figure 6a), that fosters the deposition of the eroded material again on the track, allow to understand that the depths of the wear tracks in them (Figure 9) are not higher.

Finally, in this study, the gloss and color parameters for each coating and exposure time (0 and 500 h) were also evaluated. In addition to the barrier they create between the metal being protected and the environment, the aesthetic properties of coatings are important for each application and occasion.

In Figure 12, the data corresponding to the gloss of all the PA11 coatings before (0 h) and after (500 h) the exposure to the xenon lamp are observed. Regarding the effect that nanoparticles, there is a slight tendency to decrease gloss as nanoparticles are added, although this difference can be only noticeable for 3%. Some authors have already reported that nanoparticles can decrease the initial gloss of organic coatings [11,59], although it can also be related to a slight variation in the texture and roughness of the coating.

Once the coatings have been exposed to the xenon irradiation, 3% is the only coating that slightly reduces its gloss compared to 0 h. This decrease may be due to the fact that the 3% coating surface seem to be the most damaged after xenon exposure (Figure 5). This suggests, first of all, that PA11 coatings, with and without nanoparticles, resist UV light from outside very well. Generally, the materials after being exposed to ultraviolet radiation lose gloss. This change is closely related to the polymer matrix, since for example, there are thermoset polymers that are more ultraviolet resistant as polyester [16], and less resistant, as epoxy [60]. In the case of thermoplastics polymers, they usually have higher resistance to UV [18].

Regarding the color, both a graph with the color difference (Δ*E*) and the three parameters (*L**, *a**, *b**) that define this property are shown in Figure 13 and Table 4 respectively. Δ*E* at 0 h is the color variation of each PA11 coating with reference to 0%, to know the effect of silica nanoparticles on the PA11 polymer matrix. However, at 500 h, this Δ*E* has been calculated with respect to each PA11 coating at 0 h, to analyze, in this case, the effect of the xenon exposure. 

First, Figure 13 shows a small change in color as the amount of silica nanoparticles increases (0 h). The 2% and 3% organic coatings have the same Δ*E*. Anyway, the Δ*E* is always less than 1.2, and there are studies that indicate that the color change is not important or appreciable if it does not exceed the value of 10 [61]. In any case, other authors have reported bigger changes in the Δ*E* of their coatings after nanoadditions [11,62].

Regarding coatings exposed to radiation, it is observed that the higher the amount of SiO_2_, the higher the color changes with the xenon exposure. However, it is important to point out that all of them are still below 1.7. Again, PA11 coatings confirm their high ultraviolet resistance in spite the presence of reinforcements.

In order to know what are the parameters that change each time, the values of each parameter (*L**, *a**, *b**) have been included in Table 4. The *L** parameter corresponds with colors ranging from white (+) to black (−), the *a** parameter from red (+) to green (−) and the *b** parameter from yellow (+) to blue (−). The experimental dispersion of all the tabulated values is very small, always lower than 0.1. Regarding non-exposed coatings, it can be observed that by increasing the % of SiO_2_ nanoparticles, the coatings show more positives *a** and *b** parameters (more red and yellow), and the *L**. However, it is important to realize that these changes are very small and they can be considered negligible from a practical point of view.

For the exposed PA11 coatings, it should be born in mind that generally after exposing a white polymeric material to ultraviolet, it tends to turn yellow [63,64]. For the coating under study, very small color differences caused by the xenon irradiation are monitored. The obtained data suggest that as the *a** is now more negative, all the coatings should have become slightly greener, and as *b** is more negative, they have also become slightly bluer. The changes observed with exposure are completely negligible, but they allow to clearly rule out any decrease of the white intensity of the color.

Hence, the additions of nanosilica in the studied range do not affect the good aesthetic properties of PA11 and, if its amount is lower than 2%, the durability of gloss under xenon irradiation will not be affected. The fact that the mechanical properties and wear resistance after and before irradiation are clearly better for the 1% than for 0% coating confirms the interest of manufacturing powders of PA11 with small amount of nanosilica particles (about 1%) to be electrostatically applied on metal substrates and increase the durability of its protection.

## 4. Conclusions

The main conclusions that can be drawn from the research performed in this work are:
The addition of SiO_2_ nanoparticles to powder polyamide promotes the hardening and stiffening of the coatings manufactured, while slightly reducing their crystallinity;The wear behavior of studied coatings implies an abrasive mechanism, with material removed and bonded again to the PA11 coatings, due to their plasticity;Xenon exposure does not promote important chemical changes in the coatings. Only the formation of hard imides and carboxylic acid groups has been suggested by the obtained FTIR-ATR results;After xenon exposure, 1% coating increases its hardness and stiffness, showing the lowest loss of crystallinity. This material presents the best wear behavior after irradiation, unlike the other organic coatings under study;All the studied polyamide-based coatings present excellent aesthetic properties, with very small changes in gloss (except for 3%) and color after 500 h of exposure to the xenon radiation.

## Figures and Tables

**Figure 1 polymers-12-02298-f001:**
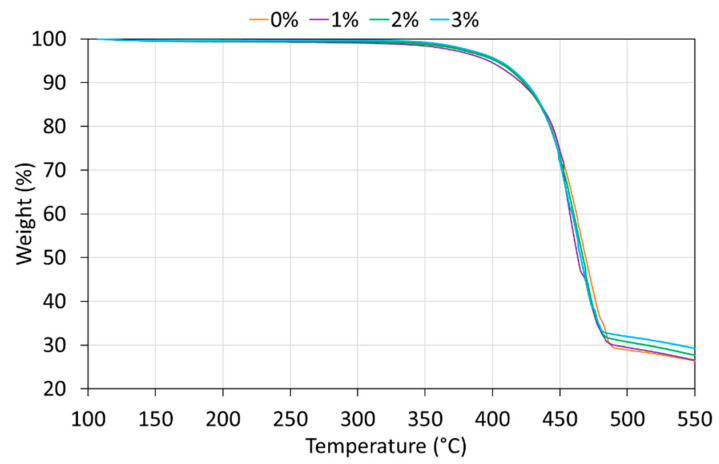
TGA curves of the non-exposed coatings.

**Figure 2 polymers-12-02298-f002:**
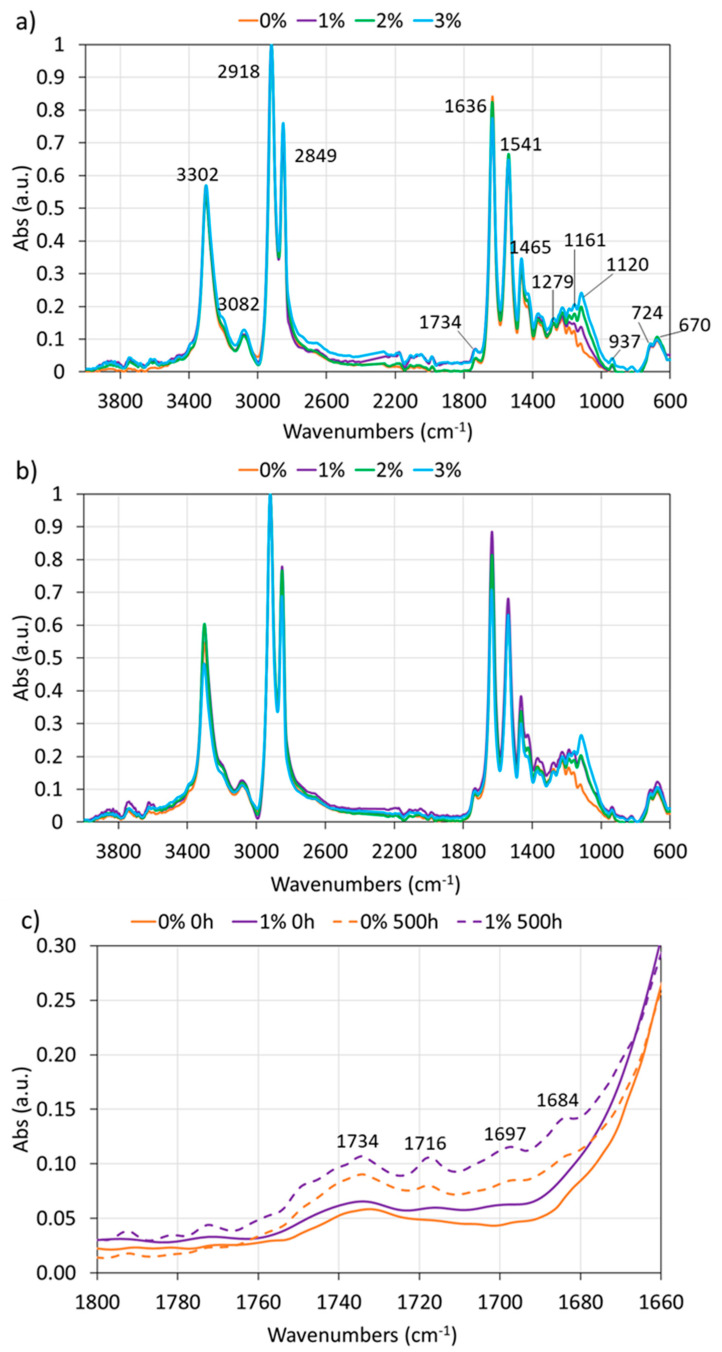
FTIR-ATR spectra of all PA11 coatings at (**a**) 0 h and (**b**) 500 h of xenon exposure, and (**c**) a detail (in the range of 1800–1660 cm^−1^) of the 0% and 1% coatings before and after xenon exposure.

**Figure 3 polymers-12-02298-f003:**
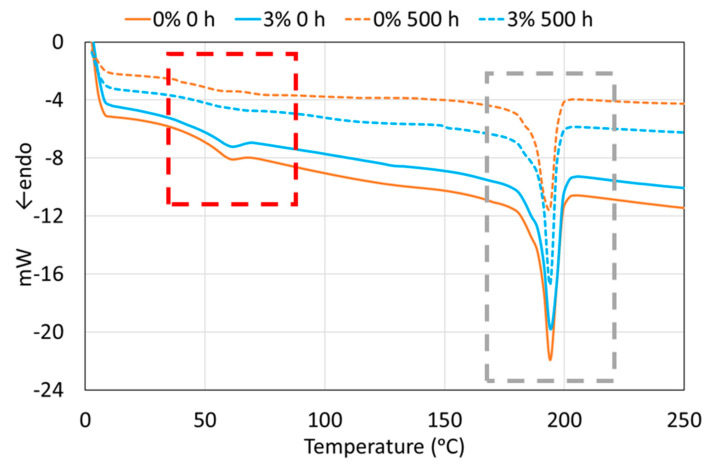
Non-isothermal DSC curves at 0 h and 500 h xenon exposure of 0% and 3% coatings.

**Figure 4 polymers-12-02298-f004:**
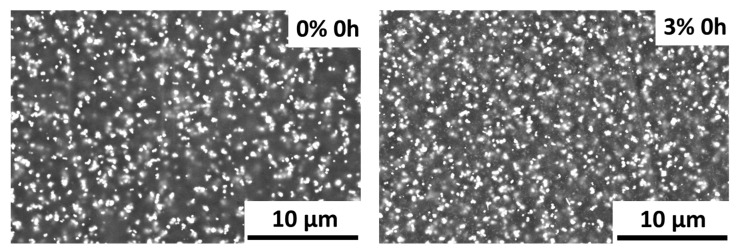
SEM micrographs of the surface 0% and 3% organic coatings before exposure (magnification ×4000).

**Figure 5 polymers-12-02298-f005:**
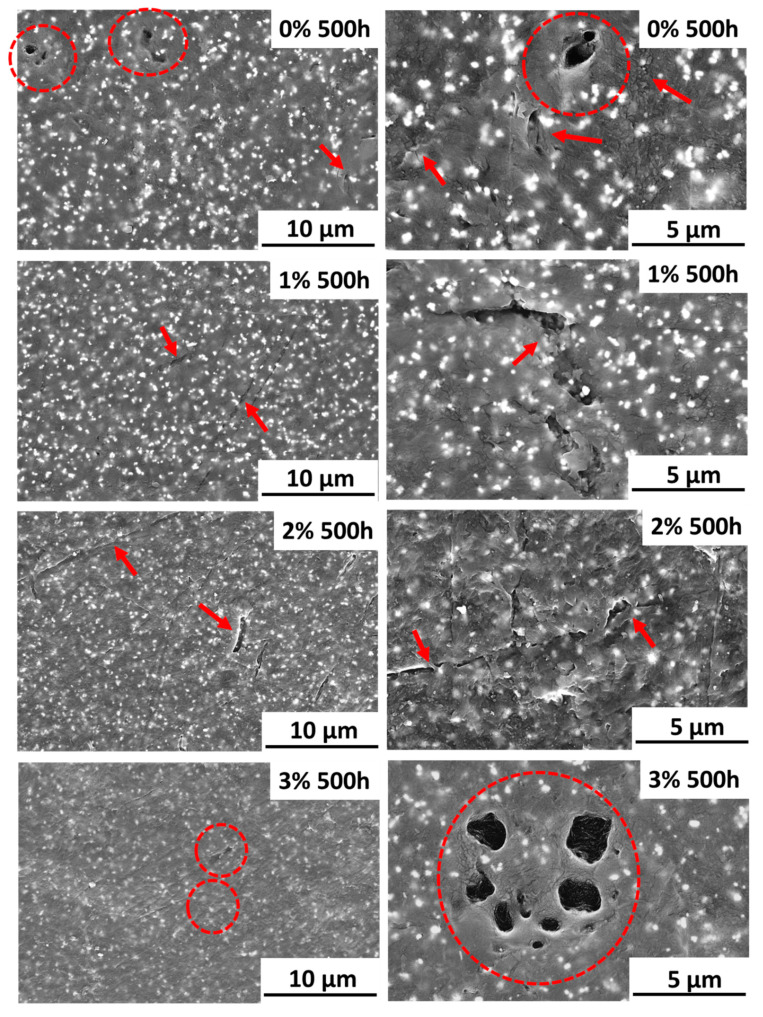
SEM micrographs of the surface of the organic coatings under study after xenon exposure (magnification of left images ×4000, right images ×8000).

**Figure 6 polymers-12-02298-f006:**
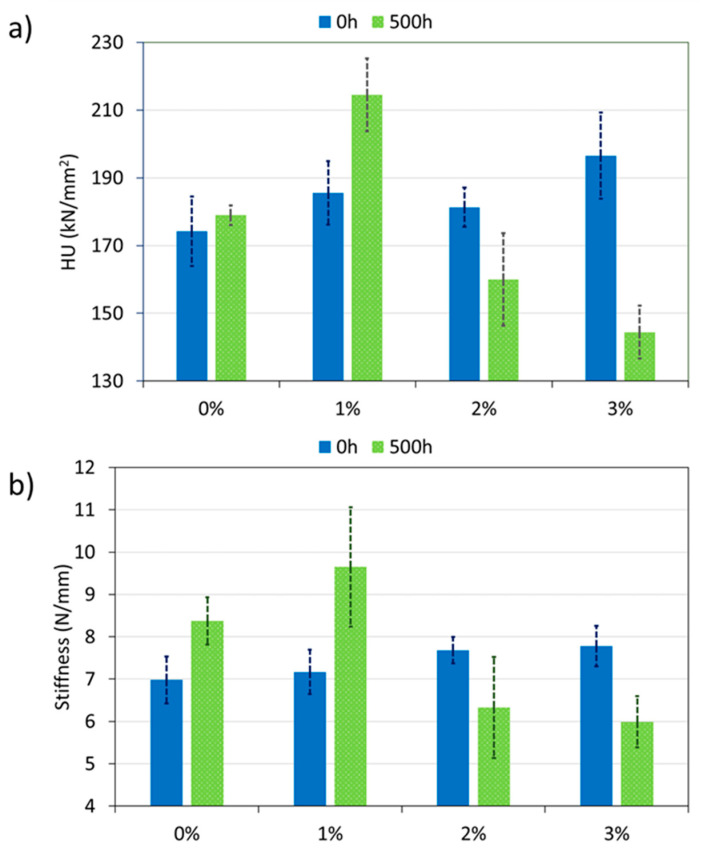
(**a**) Universal hardness (HU) and (**b**) stiffness of all organic coatings at 0 and 500 h of exposure.

**Figure 7 polymers-12-02298-f007:**
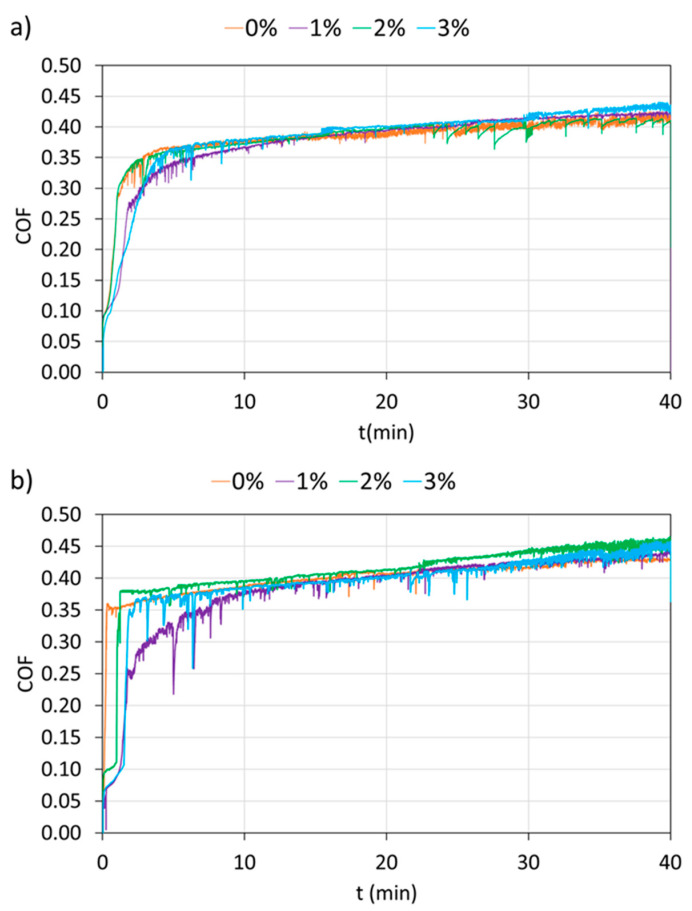
COF evolution of all organic coatings at (**a**) 0 h and (**b**) 500 h of exposure.

**Figure 8 polymers-12-02298-f008:**
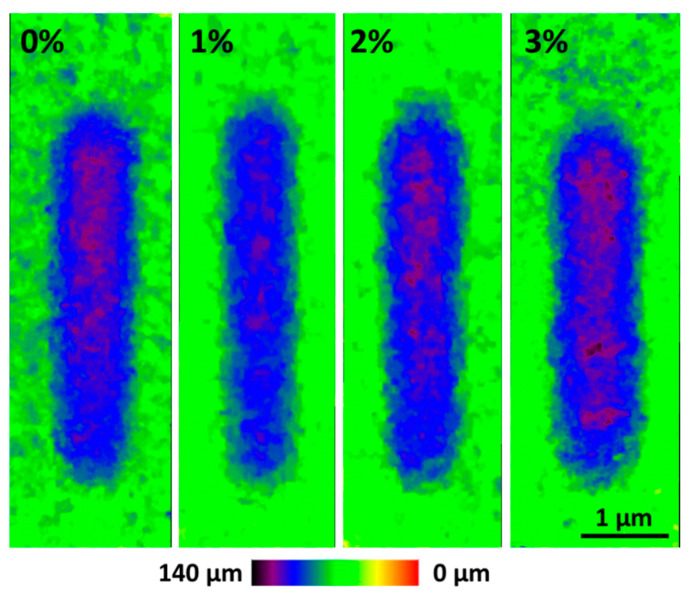
Wear tracks of all organic coatings after 500 h of xenon exposure.

**Figure 9 polymers-12-02298-f009:**
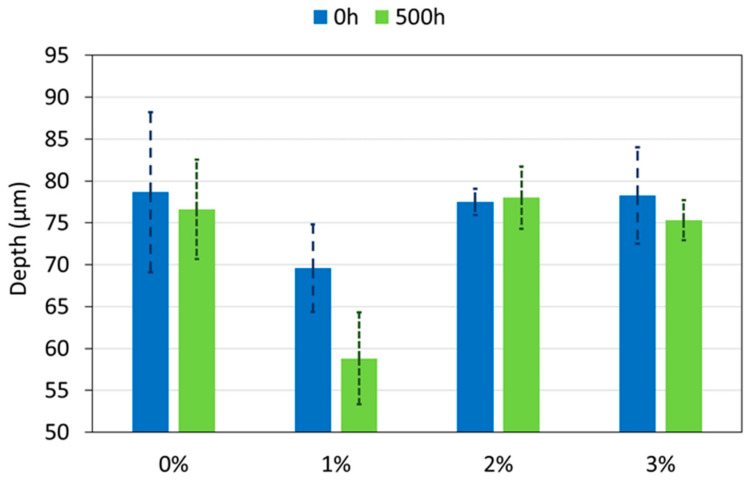
Depth of the wear tracks of all organic coatings at 0 and 500 h of exposure.

**Figure 10 polymers-12-02298-f010:**
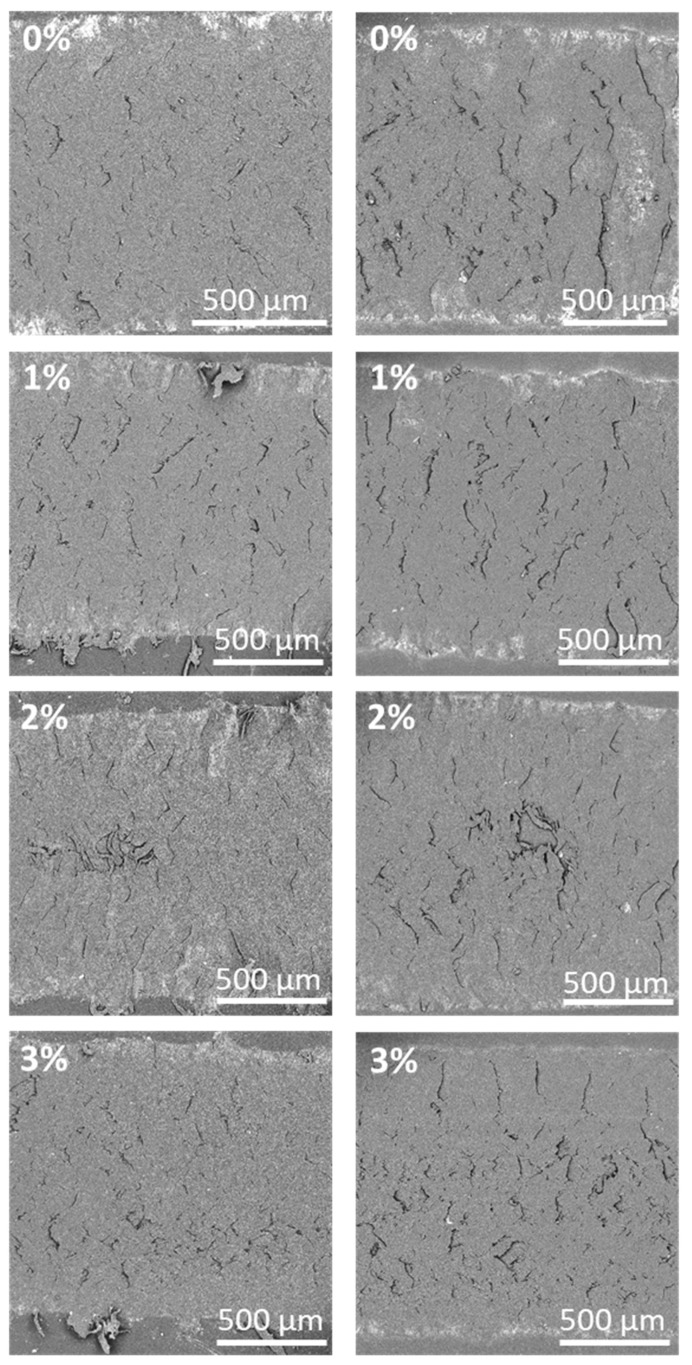
SEM micrographs of the wear mechanism of all PA11 coatings at 0 h (**left**) and 500 h (**right**) of exposure (magnification ×60).

**Figure 11 polymers-12-02298-f011:**
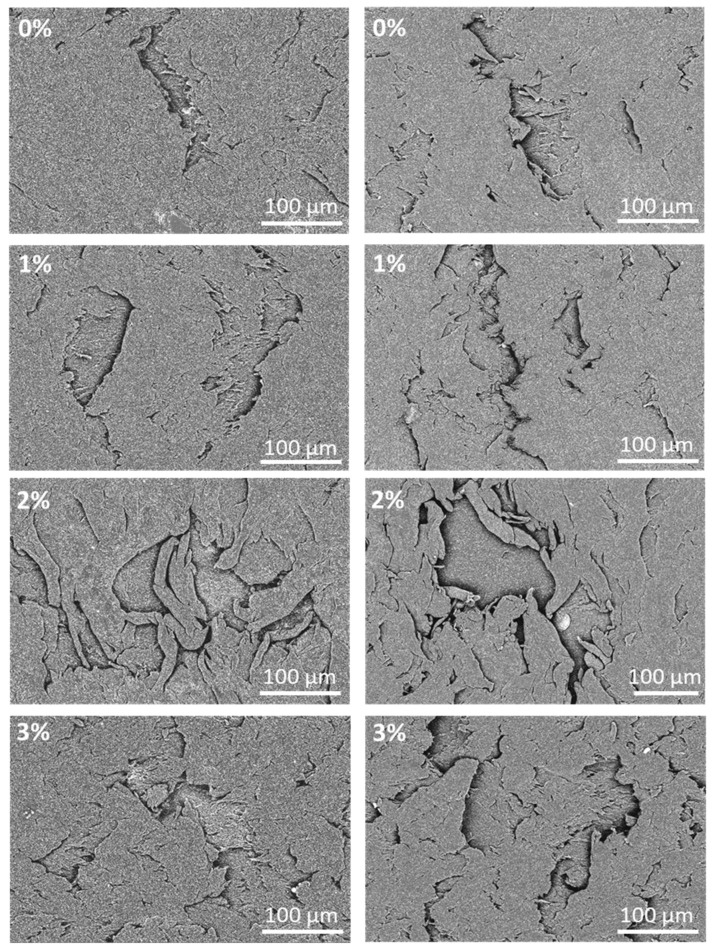
SEM micrographs showing the details of wear damages of all PA11 coatings at 0 h (**left**) and 500 h (**right**) of exposure (magnification ×300).

**Figure 12 polymers-12-02298-f012:**
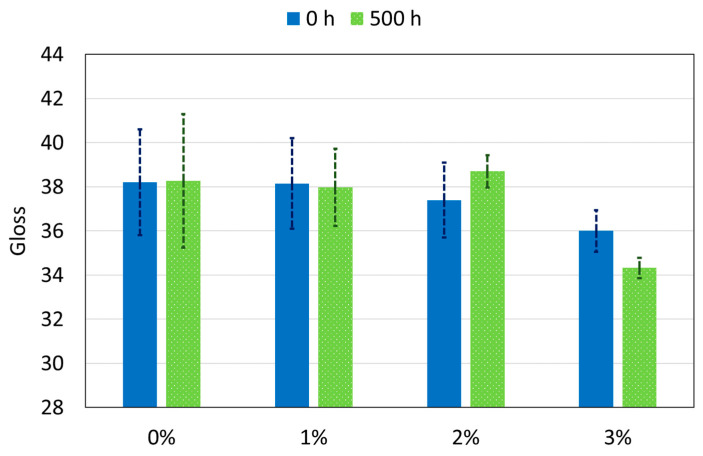
Gloss (60°) of all organic coatings at 0 h and 500 h of xenon exposure.

**Figure 13 polymers-12-02298-f013:**
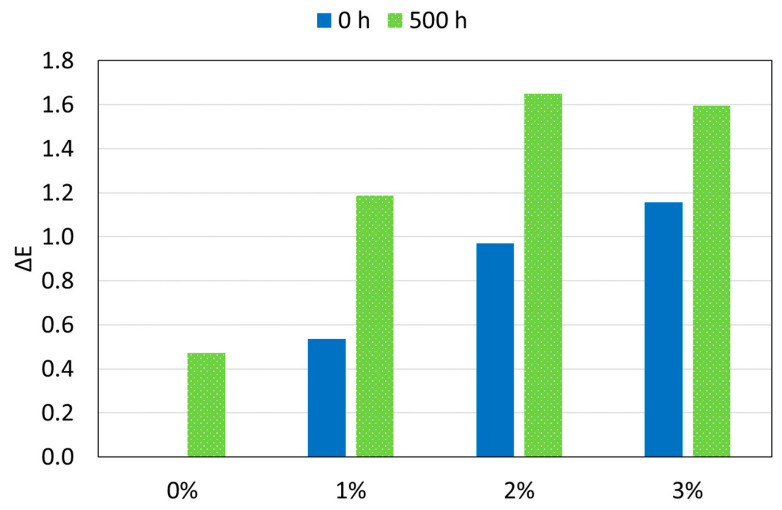
Color differences (Δ*E*) as manufactured (blue columns related to 0%) and after 500 h exposure at xenon lamp (green-related to its value at 0 h) of all organic coatings.

**Table 1 polymers-12-02298-t001:** IR absorption bands assignations [24,39,40,41].

Wavenumbers (cm^−1^)	Assignation
3302	N–H stretch
3082	N–H (Amide II)
2918	Asymmetric CH_2_ stretch
2849	Symmetric CH_2_ stretch
1734	Imides and impurities
1636	C=O stretch (Amide I)
1541	N–H and C–O (Amide II)
1465	CH_2_ scissoring vibration
1279	C–N–H (Amide III)
1161	O=C–N stretch
1114	Si–O–Si
937	Amide IV
724	CH_2_ rocking
670	Amide V

**Table 2 polymers-12-02298-t002:** Thermal properties of the organic coatings before xenon exposure.

PA11 Coatings (0 h)	*T*_g_ (°C)	*T*_m_ (°C)	Δ*H*_m_ (J·g^−1^)	*X*_C_ (%)
0%	48.7 ± 0.8	190.4 ± 1.3	27.3 ± 0.4	17.2 ± 0.2
1%	50.4 ± 0.4	190.2 ± 0.6	25.1 ± 0.3	16.1 ± 0.2
2%	48.3 ± 0.6	190.5 ± 1.8	23.9 ± 0.3	15.5 ± 0.2
3%	49.1 ± 0.4	191.1 ± 0.4	24.0 ± 0.4	15.8 ± 0.3

**Table 3 polymers-12-02298-t003:** Thermal properties of the organic coatings after xenon exposure.

PA11 Coatings (500 h)	*T*_g_ (°C)	*T*_m_ (°C)	Δ*H*_m_ (J·g^−1^)	*X*_C_ (%)
0%	40.6 ± 1.6	191.9 ± 0.4	20.6 ± 1.6	12.6 ± 0.5
1%	46.9 ± 0.4	192.7 ± 0.1	24.5 ± 0.6	15.4 ± 0.1
2%	45.2 ± 0.2	193.5 ± 0.1	22.7 ± 0.9	14.7 ± 0.6
3%	45.4 ± 0.8	193.3 ± 1.1	23.2 ± 0.6	15.3 ± 0.4

**Table 4 polymers-12-02298-t004:** Color parameters (*L**, *a**, *b**) of each PA11 coating at 0 h and 500 h of xenon exposure.

	0 h			500 h		
Sample	*L**	*a**	*b**	*L**	*a**	*b**
0%	93.5	−1.2	0.4	93.6	−1.5	0.1
1%	93.5	−1.1	0.9	94.3	−1.5	0.1
2%	93.7	−0.9	1.3	94.6	−1.5	0.2
3%	93.4	−0.6	1.4	94.2	−1.5	0.4

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
