# Peer review of "Manufacturing and Characterization of Coatings from Polyamide Powders Functionalized with Nanosilica"

_polymers, 2020, doi:10.3390/polym12102298_

Round 1
Reviewer 1 Report
Polymers
Manuscript ID939954
Manufacturing and characterization of coatings from polyamide powders functionalized with nanosilica
The present work describes the preparation of Polyamide 11 wearable coatings via melt-extrusion and compression molding of PA11 and nano-silica filler used at different loadings. In general, the work is well-organized and clear but it is complete only when the authors discuss the preparation of PA11-based coatings; in the case of PA11-silica extrusion presentation and discussion several methodological mistakes are present. I think that this manuscript should be published in Polymers only after major revision
- Line 76-81, melt extrusion process: the authors say that the filler dispersion method is important in order to ensure a good distribution of nano-silica powder and they report that melt extrusion process is the best one. Generally, researchers compare in situ polymerization technique with melt extrusion process and commonly the in situ polymerization approach is the best one (for example see Polymers 2020, https://doi.org/10.3390/polym12010211). Here, only the use of a pre-formed polymer justifies the use of melt extrusion technique. The authors should rewrite this paragraph according to this remark.
- Before melt extrusion process, why the authors do not pre-dried PA11 pellets and silica powder? The moisture content inside polymer and additive is an important parameter that can affect the real content of filler and its distribution inside polymer matrix.
- Xenon lamp: real xenon lamp irradiating power and distance between lamp and sample data are missing.
- The authors do not discuss the real content of filler after extrusion process, for example via TGA analyses. How they can demonstrate the real presence of silica and melt extrusion efficiency?
- DSC thermal program: why the authors do not consider the thermal history of the polymer? In other words, they check the composite thermal properties in first scansion…in general, the first scansion (for example 25-250°C/ isotherm of some minutes at 250°C/ cooling) is able to delete the thermal history of the polymer, a second heating allows to consider the real thermal properties of the polymer…..in this moment this section should be completely revised.
- Aging time: the authors should justify the choice of 500h, indicate a correlation between real solar exposure time and 500 h of xenon lamp and compare they results with the timeline described in reference 39.
- Figure 2: a comparison between each sample (for example neat sample before and after ageing) is more representative than the comparison between two groups of samples.
- The choice to use nano-silica loadings adopted should be motivated..in general fillers are used in high quantity….the low amounts employed here are an important advantage that is not underline in the paper.
- SEM: the authors do not discuss the distribution of silica filler in PA11 pellets after melt extrusion; how they can demonstrate the absence of aggregates, typical of such kind of process? SEM is a good way….furthermore, a comparison between neat and composite pellets should be added.
- SEM scale bar should be revised, the use of arrows should be avoided, a common rectangular bar scale seems the best option.
Author Response
The present work describes the preparation of Polyamide 11 wearable coatings via melt-extrusion and compression molding of PA11 and nano-silica filler used at different loadings. In general, the work is well-organized and clear but it is complete only when the authors discuss the preparation of PA11-based coatings; in the case of PA11-silica extrusion presentation and discussion several methodological mistakes are present. I think that this manuscript should be published in Polymers only after major revision
Line 76-81, melt extrusion process: the authors say that the filler dispersion method is important in order to ensure a good distribution of nano-silica powder and they report that melt extrusion process is the best one. Generally, researchers compare in situ polymerization technique with melt extrusion process and commonly the in situ polymerization approach is the best one (for example see Polymers 2020, https://doi.org/10.3390/polym12010211). Here, only the use of a pre-formed polymer justifies the use of melt extrusion technique. The authors should rewrite this paragraph according to this remark.
We have added a sentence in lines 81-83 with the reference, to remark the interest of in-situ polymerization.
Before melt extrusion process, why the authors do not pre-dried PA11 pellets and silica powder? The moisture content inside polymer and additive is an important parameter that can affect the real content of filler and its distribution inside polymer matrix.
In this case, we do not previously dry the materials because the PA11 was sealed and the nanosilica was stored in a container that was under vacuum. In addition, a previous DSC of the PA11 was carried out and no absorbed moisture was found. This point has been clarified in line 95 in the text.
Xenon lamp: real xenon lamp irradiating power and distance between lamp and sample data are missing.
We have indicated the distance between lamp and sample in the text (line 131-132). The real irradiating power has also been included in the text (lines 129-130).
The authors do not discuss the real content of filler after extrusion process, for example via TGA analyses. How they can demonstrate the real presence of silica and melt extrusion efficiency?
We agree with the reviewer. We checked through TGA of the coatings that everything was correct after extrusion and coating formation. We have included a graph (Figure 1) in order to demonstrate the presence of silica and the efficiency of the extrusion process. The information about TGA tests has been included in the experimental part (lines 125-127)
DSC thermal program: why the authors do not consider the thermal history of the polymer? In other words, they check the composite thermal properties in first scansion…in general, the first scansion (for example 25-250°C/ isotherm of some minutes at 250°C/ cooling) is able to delete the thermal history of the polymer, a second heating allows to consider the real thermal properties of the polymer…..in this moment this section should be completely revised.
We agree with the reviewer about the importance of the thermal history of the polymers in many cases. We have preferred to focus on the as-manufactured coatings, for considering that it would be a condition similar to their eventual in-service use. Anyway, for being sure about this fact we were aware to, we performed two scans and we checked that there were no significant changes between them (no thermal history that could affect the final results). This point has been clarified in lines 142-143.
Aging time: the authors should justify the choice of 500h, indicate a correlation between real solar exposure time and 500 h of xenon lamp and compare they results with the timeline described in reference 39.
Regarding the choice of 500 h in this work, in can be included among the typical UVB time exposure range that can be found in the literature. We started with 250 h and then we decided to increase to 500 h of xenon exposure, where we could already see some differences. Then, we gave priority to studying all the properties instead of increasing the exposure.
Though it would be really interesting to propose a correlation between the performed test and real exposure, it is really difficult to drawn it in a reliable way. There are many experimental factors that can influence the results, such as the power, the distance, the source, etc. This correlation would be influenced, not only by the testing conditions, but also by the real location. Without testing in a specific location over a number of years, it is impossible to perfectly predict product response. Unfortunately, for the reference 39, we have not found a timeline that can be extrapolated to our work. We are aware that it has even proposed that 500 h of a determined UVB bulbs exposure can be compared to 1 year of exposure in Florida, but the use of those bulbs has declined due to their poor accuracy predicting outdoor performance [Weathering Testing Guidebook, Atlas, 2001]. However, we consider risky to extrapolate this information to our testing conditions. The validation of the results with outdoor testing exposure is an interesting further step for fully optimized coatings.
Figure 2: a comparison between each sample (for example neat sample before and after ageing) is more representative than the comparison between two groups of samples.
We have considered your suggestion and we have changed the old Figure 2 (new Figure 3).
The choice to use nano-silica loadings adopted should be motivated..in general fillers are used in high quantity….the low amounts employed here are an important advantage that is not underline in the paper.
Thank you very much for the comment, we have underlined this fact in the manuscript (lines 95-97). Moreover, the results prove that amounts higher than 2% are not positive for the conditions under study.
SEM: the authors do not discuss the distribution of silica filler in PA11 pellets after melt extrusion; how they can demonstrate the absence of aggregates, typical of such kind of process? SEM is a good way….furthermore, a comparison between neat and composite pellets should be added.
We have carried out the SEM observations after coating formation, which we considered the key determining step for the in-service performance of the coatings. It is true that an observation of the pellets could have been also interesting. Unfortunately, we have run out for the pellets now. However, we have assumed that the good results in the final step can very probably be a consequence of a good mixing process.
SEM scale bar should be revised, the use of arrows should be avoided, a common rectangular bar scale seems the best option.
Thanks for the recommendation; we have changed all SEM scale bars.
Reviewer 2 Report
This paper studied the polyamide coatings functionalized with nanosilica. Some minor comments are provided as follows. The results are comprehensive. Therefore, I would like to recommend it for publication upon a minor revision.
The topic of this paper is about coatings; however, the organic coatings were made by compression molding, which seems unrealistic to make conformal coatings onto the base materials. Please make it clear.
The explanation of the reduction of mechanical and tribological properties should be strengthen.
Some errors are found with the citations.
Author Response
This paper studied the polyamide coatings functionalized with nanosilica. Some minor comments are provided as follows. The results are comprehensive. Therefore, I would like to recommend it for publication upon a minor revision.
The topic of this paper is about coatings; however, the organic coatings were made by compression molding, which seems unrealistic to make conformal coatings onto the base materials. Please make it clear.
We agree with the reviewer. The application of powder coatings is usually done by electrostatic application. However, in this case, once the materials are mixed in the extruder, pellets are obtained, so the powder would have to be remanufactured to be able to apply electrostatically. To do this, we should make a cryogenic grinding. In this work, we seek to optimize the amount of nanoreinforcement in order to grind and apply the optimized materials through electrostatic application in the future. We have added a comment about it in the manuscript (lines 109-114), added the word “electrostatically” to clarify the sentence in line 434.
The explanation of the reduction of mechanical and tribological properties should be strengthen.
We have reinforced the explanation about this (lines 296-300, 341-342).
Some errors are found with the citations.
We have checked all citations and we have corrected all errors.
Round 2
Reviewer 1 Report
The authors carefully revised the manuscript, adding relevant data according to my suggestions and replying point by point allo of my observations. I think the current version of the manuscript is fine for Polymers.